:ּ:PLOS | ONE

# Effectiveness of different central venous catheter fixation suture techniques: An *in vitro* crossover study

**Manuel Florian Struck**[1☯*], **Lars Friedrich**[1☯¤a], **Stefan Schleifenbaum**[2¤b],
**Holger Kirsten**[3], **Wolfram Schummer**[4¤c], **Bernd E. Winkler**[5]

**1** Department of Anesthesiology and Intensive Care Medicine, University Hospital Leipzig, Leipzig, Germany,
**2** ZESBO-Zentrum zur Erforschung der Stütz- und Bewegungsorgane, Leipzig, Germany, **3** Institute for
Medical Informatics, Statistics, and Epidemiology (IMISE), University of Leipzig, Leipzig, Germany,
**4** Department of Anesthesiology and Intensive Care Medicine, University Hospital Jena, Jena, Germany,
**5** Department of Anesthesiology and Intensive Care Medicine, University Hospital Würzburg, Würzburg,
Germany

☯ These authors contributed equally to this work.
¤a Current address: Department of Anesthesiology, Leipzig Heart Center, Leipzig, Germany
¤b Current address: Department of Orthopedic, Trauma and Plastic Surgery, University Hospital Leipzig,
Leipzig, Germany
¤c Current address: Department of Anesthesiology and Intensive Care Medicine, HELIOS Spital Überlingen,
Überlingen, Germany
* manuelstruck@web.de

Catanzaro, ITALY

**Data Availability Statement:** All relevant data are
within the paper and its Supporting Information
files.

## Abstract

### Purpose

Proper fixation of central venous catheters (CVCs) is an integral part of safety to avoid dis-
lodgement and malfunction. However, the effectiveness of different CVC securement
sutures is unknown.

### Methods

Analysis of maximum dislodgement forces for CVCs from three different manufacturers
using four different suture techniques in an in vitro tensile loading experiment: 1. "clamp
only", 2. "clamp and compression suture", 3. "finger trap" and 4. "complete", i.e., "clamp +
compression suture + finger trap". Twenty-five tests were performed for each of the three
CVC models and four securement suture techniques (n = 300 test runs).

### Results

The primary cause of catheter dislodgement was sliding through the clamp in techniques 1
and 2. In contrast, rupture of the suture was the predominant cause for dislodgement in
techniques 2 and 3. Median (IQR 25–75%) dislodgement forces were 26.0 (16.6) N in tech-
nique 1, 26.5 (18.8) N in technique 2, 76.7 (18.7) N in technique 3, and 84.8 (11.8) N in tech-
nique 4. Post-hoc analysis demonstrated significant differences *(P < .001)* between all
pairwise combinations of techniques except technique 1 vs. 2 *(P = .98)*.

**Funding:** The authors received no specific funding for this work.

**Competing interests:** The authors have declared that no competing interests exist.

## Conclusions

"Finger trap" fixation at the segmentation site considerably increases forces required for dislodgement compared to clamp-based approaches.

## Introduction

Central venous catheterization is a common procedure to provide safe administration of vasoactive agents, fluid resuscitation, hemodialysis, and hemodynamic monitoring [1,2]. Based on numerous studies, recommendations have been published regarding CVC placement [3–5]. Although there is increasing evidence for how to increase safety regarding CVC placement, studies regarding CVC fixation are scarce. The American Society of Anesthesiologists (ASA) recommends suturing the CVC to the patient's skin with no mention of a particular technique [5]. The effectiveness of different CVC fixation sutures is unknown [6]. Moreover, sutureless CVC fixation devices have recently become increasingly popular because of their less invasive nature and potentially greater patient comfort [7–11]. However, CVC fixation with sutures is still widely used and mandatory in high-risk patients (i.e., burns). Due to individual experiences of the authors CVC dislodgements are rare when utilizing a "finger trap"-based fixation and more common when relying on the "clamp only" but until now, experimental data that support these observations are not available.

In the present study, we therefore evaluated the strengths of different securement suture techniques under standardized in vitro conditions using mechanical tension forces. We hypothesized that a higher complexity fixation suture would be associated with increased strength of the CVC-suture system and thus increased safety to prevent CVC dislodgement. The primary aim of this study was to compare the stability of four different suture techniques, each applied to CVC models from three different manufacturers. The secondary aim was to analyze possible mechanical damage to the CVC during tensile loading.

## Methods

The CVCs were sutured on a 5 x 5 cm piece of untanned cow leather tissue (thickness: 1.2–1.4 mm) with a central perforation (standardized using a 2 mm belt punch) for CVC insertion. All pieces of leather were obtained from a single animal. In our preliminary tests, the tear strength of the leather was determined to fall in a range of 500–700 N. One single practitioner (board certified specialist in anesthesiology and intensive care medicine) sutured all catheters in a standardized manner. We used a total of 300 catheters total provided by Arrow®/Teleflex®, Wayne, USA (30 cm, 7 Fr./2.4 mm: 1x16 G, 2x18 G), B. Braun, Melsungen, Germany (Certofix® Trio 730, 30 cm, 7 Fr./2.4 mm: 1x16 G, 2x18 G), and Vygon, Écouen, France (MultiCath 3™, 20 cm, 7,5 Fr/2.7 mm: 1x14 G, 2x18 G). For the sake of simplicity, the terms "Arrow", "B. Braun" and "Vygon" will be used consistently below. In a preliminary test run, we evaluated the median tear strength of the CVCs ($N = 5$ per manufacturer; Arrow 83.9 N, B. Braun 70.1 N, and Vygon 98.3 N, respectively). Mersilene® 2.0 (Ethicon, Johnson-Johnson, Norderstedt, Germany) sutures were used for fixation by tying three knots. All catheters were secured within a distance of 5 cm from the perforation to the segmentation site to compensate for differing catheter lengths.

1. Standard clamp ("clamp only"), the most commonly used fixation approach: The catheter was attached via two single sutures in each of the holes of the clamp [Fig 1A].

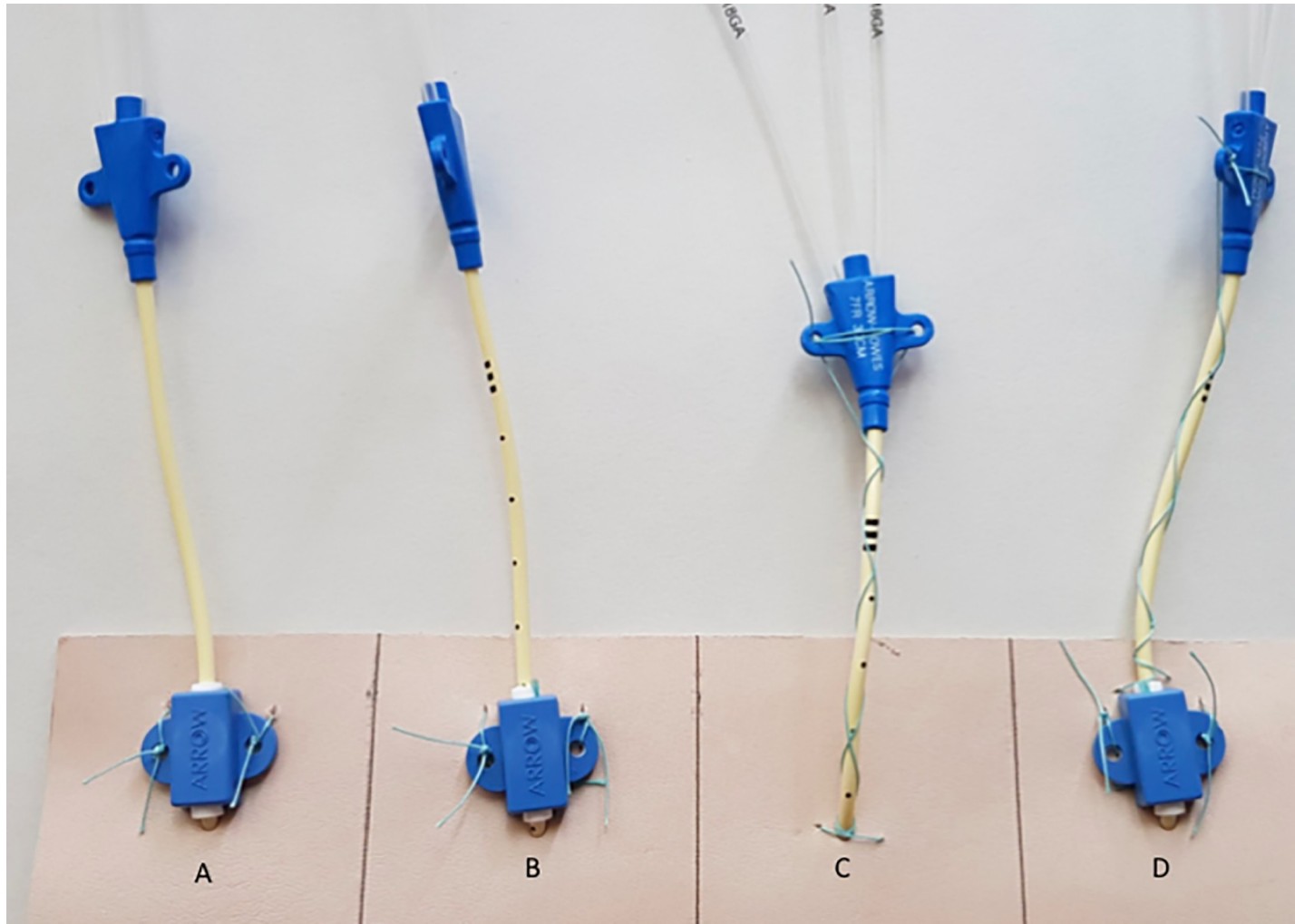

**Fig 1. Fixation techniques.** From the left to the right, the four fixation techniques used in the present study are presented (A-D). A represents the "standard" technique, B represents the "compression" technique, C represents the "finger trap" technique and D represents the "complete" technique. Details of the compression technique are presented in Fig 2.

2. Standard clamp + compression sutures ("compression"), a measure to increase friction to the rubber part of the clamp: The soft inner part of the compression clamp was compressed towards the catheter via two additional compression sutures [Fig 1B, Fig 2].

3. Finger trap ("finger trap"), A potential approach of fixation in cases of accidental loss or damage of the fixation clamp: The catheter was attached to the leather via a single stitch at the puncture site, and the two ends of the suture were wrapped around the catheter like a finger trap. The ends of the suture were tied together at the segmentation site of the lumens [Fig 1C].

4. Complete fixation ("complete"). The combination of the approaches 1+2+3 to achieve optimal stability: The "clamp only", "compression", and "finger trap" approaches were combined [Fig 1D].

For all experiments, a tensile testing apparatus (Type 5566A, Instron INC, Norwood, MA, USA) was used at a biomechanical laboratory. Parts and mounting of the tensile testing apparatus were manufactured and specifically modified for the present study in order to provide optimal comparability and test reliability. Forces were measured in Newtons (1 N = 0.102 kg).

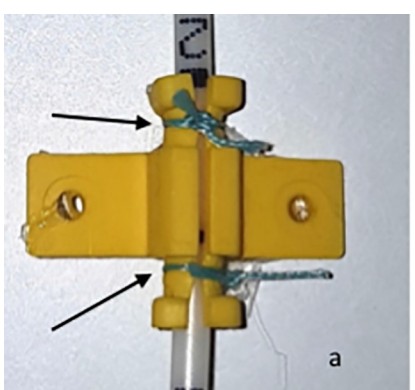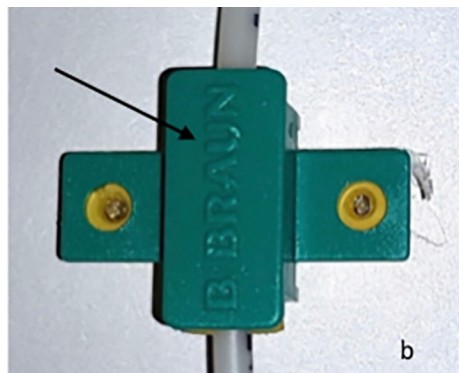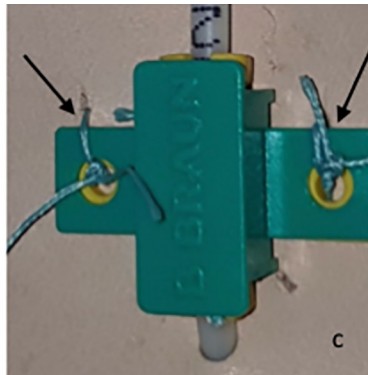

**Fig 2. Steps of the "compression" technique. A)** Two sutures are placed at the inner soft part of the clamp to increase compression of the soft part and to increase friction between the soft part of the clamp and the catheter. **B)** The outer rigid part of the clamp is placed on top of the soft part. **C)** The clamp is sutured to the leather.

The leather part was attached to a screw compressor clamp at the static part of the testing apparatus, while the segmentation site of the CVC was mounted to the dynamic part via a mechanical fastening unit, which was specifically designed and evaluated for our experiments [Fig 3A and 3B]. The distance between the static and the dynamic parts was identical in each run. The traction speed of the testing apparatus was set to 1,000 mm min$^{-1}$, and a precise displacement-force-curve was recorded via the integrated "Bluehill 2" software (Instron INC, Norwood, MA, USA). The distance of CVC dislodgement was recorded electronically, and dislodgement of 5 cm at the insertion site was considered critical in this test setting. The maximum force required for catheter dislodgement was compared among the four catheter fixation techniques. Suture ruptures, CVC ruptures, and other events were recognized and recorded. Twenty-five test runs were performed for each combination of the three manufacturers and four securement techniques, resulting in a total of 300 test runs.

## Statistical analysis

After exporting data from the "Bluehill 2" software of the testing apparatus, statistical analysis was performed using Microsoft Excel Mac 2016 (Microsoft Inc, Redmond, WA, USA) and SPSS 25 (IBM Inc, Armonk, NY, USA). Testing for normal distribution was performed using the Kolmogorov–Smirnov Test. After testing for normal distribution, the effect of fixation technique and manufacturer was analyzed utilizing two-way factorial analysis of variance (ANOVA). Post-hoc analysis of the ANOVA results was performed for pairwise comparisons of individual fixation approaches and manufacturers. *P*-values were adjusted for multiple testing via Bonferroni correction. *P*-levels less than 0.05 were considered significant.

## Results

The primary cause for catheter dislodgement in the "clamp only" and "compression" groups was sliding through the clamp [Figs 4 and 5, S1 Fig]. In contrast, the primary cause of dislodgement for the "finger trap" and "complete" fixation groups were rupture of the fixation suture [Figs 4 and 5, S1 Fig]. In all manufacturers, the finger-trap based fixation techniques "finger trap" and "complete" resulted in a substantial increase in dislodgement forces compared to "clamp only" and "compression" [Fig 5]. The dislodgement forces using the "complete" fixation technique was 6.7 times as high as using the "clamp only" fixation technique (median 87.7 *vs.* 13.1 N, $P < 0.01$) in Arrow CVCs, 2.8 times as high (median 84.7 *vs.* 30.7 N, $P < 0.01$) in B.

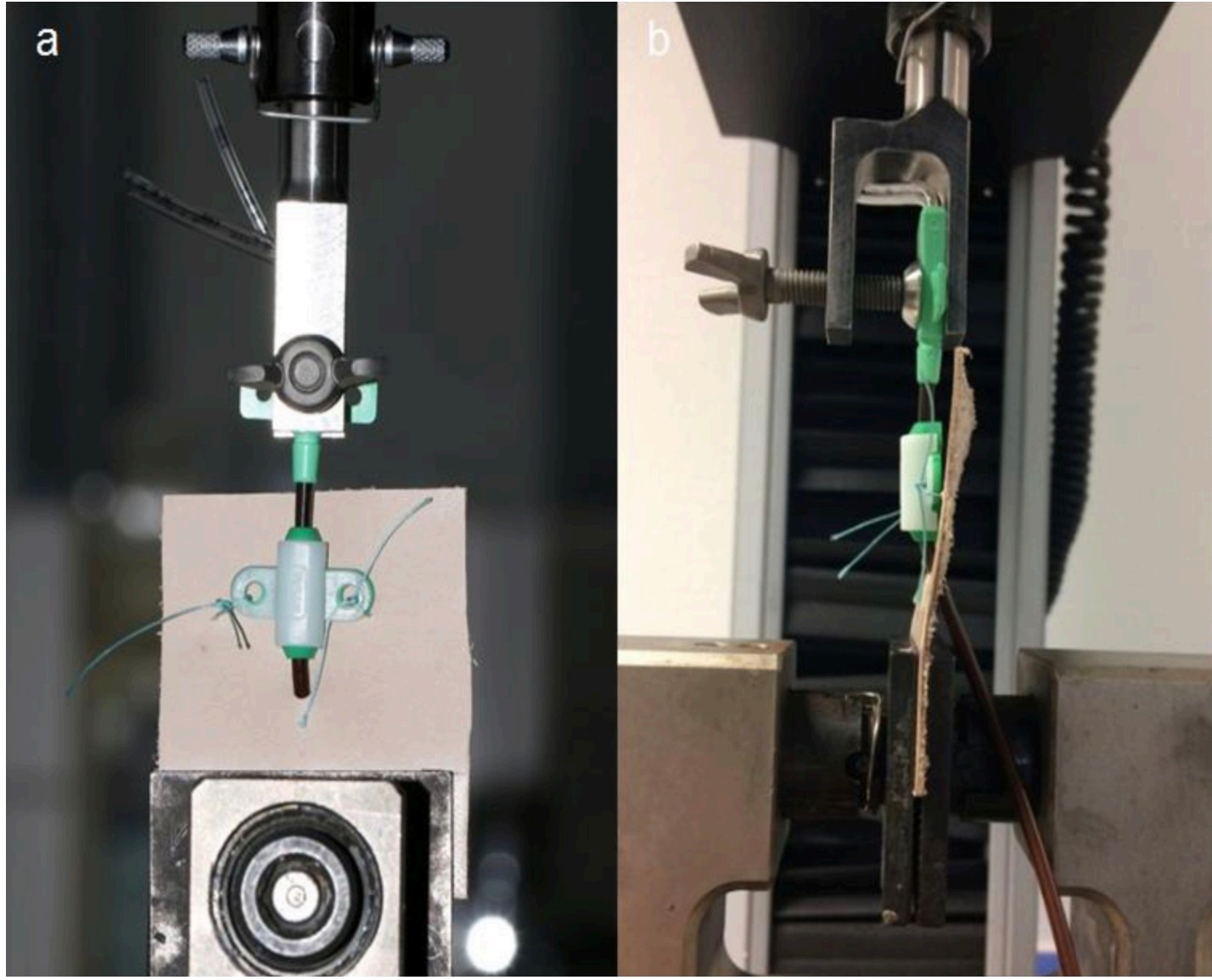

**Fig 3. Test setting.** Anterior view (a) and lateral view (b) of the sutured central venous catheter unit mounted on the tensile testing machine.

Braun CVCs and 3.4 times as high (median 84.8 *vs.* 25.2 N, $P < 0.01$) in Vygon CVCs [Figs 4 and 5].

Table 1 presents group statistics of the dislodgement forces depending on the fixation technique.

According to the ANOVA results, fixation technique and manufacturer both had a significant impact on the force required for catheter dislodgement ($P < 0.001$). Post-hoc analysis demonstrated that dislodgement forces were not higher when "compression" technique was used compared to "clamp only" ($P = 0.98$). Both, "finger trap" and "complete" resulted in higher dislodgement forces than "clamp only" ($P < 0.001$ each). "Finger trap" and "complete" also resulted in higher dislodgement forces than the "compression" technique ($P < 0.001$ each). Dislodgement forces in the "complete" fixation approach were significantly ($P < 0.01$) higher than in the "finger trap" approach. Furthermore, dislodgement forces were higher in B.

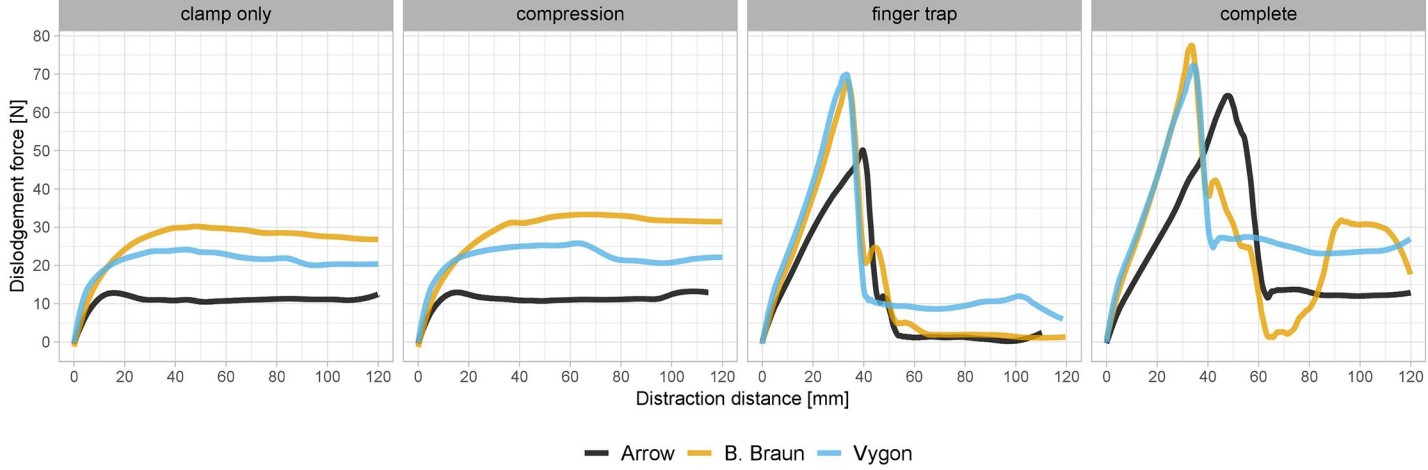

**Fig 4. Distance-force diagrams.** Median forces and distances of all investigated fixation suture techniques. A steady state of dislodgement force occurs in the "clamp only" and "compression" fixation technique. The catheter is being pulled throughout the fixation clamp in these two approaches. In contrast, the dislodgement force increases steadily in the "finger trap" and "complete" fixation technique until a rupture of the suture or catheter occurs. After the rupture, the dislodgement force is close to zero in the "finger trap" approach since no clamp is being used. In contrast, the dislodgement force in the "complete" fixation technique is similar to the "clamp only" and "compression" approach since the catheter is still secured by the clamp after the rupture of suture. The second peak observed in "finger trap" and "complete" technique of Arrow and B. Braun appears when the catheter ruptured before the suture.

Braun ($P < 0.01$) and Vygon ($P < 0.01$) CVCs compared to Arrow CVCs but did not differ significantly between B. Braun and Vygon CVCs ($P = 0.158$).

Upon examination of the individual manufacturers, dislodgement forces depended on the fixation techniques as follows:

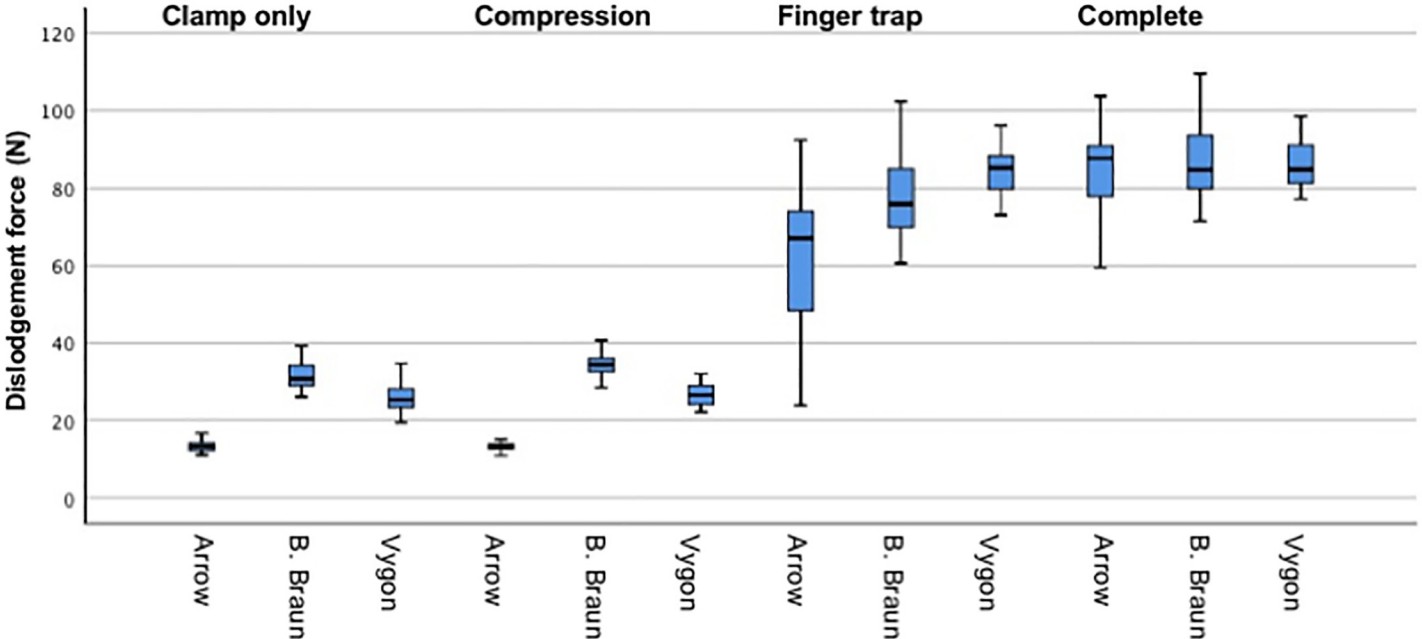

**Fig 5. Dislodgement forces.** Dislodgement forces measured in Newton (N). The boxes represent median, 25% and 75% quantile. The whiskers represent 1.5 x interquartile range. 300 test runs were performed in total, 25 test runs were performed in each experimental group.

**Table 1. Dislodgement forces depending on the fixation technique.**

|  | Clamp only (n = 75) | Compression (n = 75) | Finger trap (n = 75) | Complete (n = 75) |
|---|---|---|---|---|
| Median | 26.0 N | 26.5 N | 76.7 N | 84.8 N |
| 95% CI | 23.1–28.1 N | 22.9–29.5 N | 72.7–80.7 N | 82.6–88.1 N |
| Range | 10.9–42.4 N | 10.6–50.6 N | 23.7–102.2 N | 52.2–115.0 N |
| IQR 25%-75% | 16.6 N | 18.8 N | 18.7 N | 11.8 N |

Data are median, 95% confidence Interval (CI) of the median, minimum–maximum, and interquartile range (IQR) 25%-75%.

"Compression" technique did not increase dislodgement forces compared to "clamp only" (P = 0.98). "Finger trap" (P < 0.001) and "complete" (P < 0.001) resulted in higher dislodgement forces than "clamp only". Furthermore, "finger trap" (P < 0.001) and "complete" (P < 0.001) resulted in higher dislodgement forces than "compression". The highest dislodgement forces were observed in the "complete" fixation approach and were significantly (P < 0.01) higher than in the "finger trap" approach.

- Arrow: "clamp only" = "compression" (*P = 0.97*), "compression" < "finger trap" (*P < 0.01*), "finger trap" < "complete" (*P < 0.01*)

- B. Braun: "clamp only" = "compression" (*P = 0.24*), "compression" < "finger trap" (*P < 0.01*), "finger trap" = "complete" (*P = 0.09*)

- Vygon: "clamp only" = "compression" (*P = 0.98*), "compression" < "finger trap" (*P < 0.01*), "finger trap" = "complete" (*P > 0.96*)

During tensile loading, we observed several instances of mechanical damage to the catheters: In the "finger trap" fixation technique, six ruptures of Arrow catheters, one rupture of an Arrow eyelet and eight ruptures of B. Braun catheters occurred during tensile loading. Three ruptures of Arrow CVCs and 14 ruptures of B. Braun CVCs were observed in the "complete" fixation technique. All catheters ruptured a few millimeters distal to the segmentation site. One catheter ruptured at the leather insertion site. The median forces at which catheter rupture occurred were 55.4 N for Arrow catheters and 81.9 N for B. Braun catheters, respectively. The eyelet rupture of the Arrow catheter occurred at 68.2 N.

## Discussion

Catheter dislodgement is a frequent complication after central venous catheterization [12]. CVC dislodgement can cause extravascular drug administration and infiltration (e.g., potassium, chemotherapy, vasoactive drugs), potentially resulting in tissue necrosis. Moreover, it may lead to inappropriate or failed administration of vasopressors or inotropes resulting in severe hypotension and circulatory arrest [7–14]. Thus, securement of CVCs is essential for safety reasons [5]. Reasons for CVC dislodgement are patients transfers (e.g., to and from computed tomography or in the operating room) and auto-aggressive behaviour (e.g., hyperactive delirium). Because these scenarios are difficult to simulate in a standardized manner we developed an experimental tensile loading setup to investigate different suture techniques.

In this study, the forces required for catheter dislodgements significantly depended upon the fixation technique and the manufacturer of the CVC. The "compression" technique was not associated with higher dislodgement forces compared to the "clamp only" technique. In contrast, both the "finger trap" and "complete" fixation techniques significantly increased the dislodgement force and are likely to reduce CVC dislodgement rates in clinical practice. With regard to both inward and outward CVC dislodgement, the combination of the techniques "clamp only" plus "finger trap" might as well be a reasonable alternative to the "complete" fixation technique although this combination was not tested in the study.

Regarding individual manufacturers, "complete" fixation did result in higher dislodgement forces compared to "finger trap" fixation in Arrow catheters only. However, it has to be taken into account that the present study investigated outward dislodgement only, and it is likely that a fixation clamp or adhesive dressings are protective against inward dislodgement which can result in contamination of the insertion site, potential infections, potential damage to vessels and cardiac structures, and cardiac tamponade [15,16]. In clinical practice, some situations (i.e., loss of clamp parts) can require the combination of "finger trap" fixation and dressing adhesives for CVC securement. While "finger trap" and "complete" fixation techniques contributed to a considerable increase in dislodgement force, severe instances of mechanical damage to CVCs were observed in Arrow and especially in B. Braun but not in Vygon catheters during traction with higher forces. The additional suturing of the segmentation site has been recommended to avoid dislodgement [6], even though this approach has not been evaluated systematically. A recent clinical study demonstrated that a combination of "clamp only" plus "finger trap" was superior to "clamp only" plus "fixation of the segmentation site" with regards to dislodgement rates and catheter kinking [17]. Our data support the assumption that fixation of the segmentation site increases the dislodgement force. However, we observed that the suture may cause a manufacturer-dependent cutting effect to the catheter during tensile loading which might be a result of the test setup and does not necessarily reflect clinical practice. The higher abrasion resistance of Vygon catheters might be associated with the slightly higher diameter or different material composition (stiffness) compared to Arrow and B. Braun. Furthermore, Arrow CVCs showed a considerable variability in maximum dislodgment forces due to deformation at the segment site of the CVCs. The segmentation site of Arrow CVCs consists of soft plastic material, while Vygon and Braun CVCs use more rigid material with almost no deformation during tensile loading. Further studies should explore if this effect is rather due to experimental conditions or a clinically relevant phenomenon.

## Comparison with other fixation techniques

The "clamp only" and "compression" techniques using B. Braun and Vygon catheters resisted dislodgment forces as high as approximately 30 N that is in line with data published by Rutledge *et al.* who reported suture dislodgement forces of 28 N in a porcine model [8]. They reported slightly higher maximum forces if adhesive techniques were applied (37 N with Statlock™, 40 N with Tegaderm™ and 41 N with Tegaderm CHG™) rather than a suture technique (28 N) [8]. The Sorbaview™ adhesive, in contrast, resulted in a lower dislodgement force (17 N). Another study found a trend towards an increased rate of accidental CVC pullout when the Statlock™ adhesive approach was used compared to sutures [12] while recent data did not find statistical differences in unplanned CVC removal and CVC migration when using suture-free systems compared to sutures [11]. In two clinical studies comparing CVC sutures with a securing clamp, the authors found less time spent in clamp fixation than in suturing [18,19]; one of these studies observed superior strength than in sutures [19], whereas the other had unacceptable rates of accidental CVC pullout [18]. Another study demonstrated that a suture approach resulted in a mean ± SD dislodgement force for Arrow CVCs of 40.9 ± 10.7 N, while stapling with four 0.022 in. staples or four 0.025 in. staples resulted in dislodgement forces of 34.0 ± 7.2 N and 40.4 ± 5.8 N, respectively [20]. In contrast to our experiments, the staples and sutures in this study were placed at the segmentation site of the catheters and did not use the fixation clamps of the catheters.

## Limitations

The present study evaluated four different CVC securement suture techniques applied in three different CVC models in an experimental setting. Due to the lack of previous studies, it was

impossible to perform a precise sample size calculation prior to the experiments. However, based on the distinct significant differences between the individual fixation approaches and the low intra-group-variability, we regard the sample size as being appropriate for a pilot study. In order to avoid the ethical implications of working with live animals or humans and to provide high setup standardization, we chose cow leather tissue for the suture experiments. Our material tests revealed tear resistances of the leather of 500–700 N which are comparable with human skin providing resistances of up to 882 N [21]. The applied dislodgement force of 1,000 mm min$^{-1}$ was chosen to provide data of potential clinical situations of catheter dislodgement. Faster or slower dislodgement speed may have provided different results. The direction of the tensile loading force was 180 degrees from the mounted tissue-CVC unit, and other angles (e.g., 90 degrees) may have provided different results. We chose a critical dislodgement distance of 5 cm, which may be clinically relevant in both right-sided (risk of extravasation) and left-sided (vascular wall arrosion) thoracocervical puncture approaches [3,15]. Mersilene 2.0 sutures were used since it is the standard suture in the Departments of the authors. Other sutures may have provided different results. Furthermore, we did not apply any adhesive dressing to the CVC, which might contribute to the stability of the CVC placement in the clinical setting.

In conclusion, our results support the hypothesis that "finger trap" fixation at the segmentation site considerably increase outward dislodgement forces compared to clamp-based approaches.

## Supporting information

**S1 Dataset. Dislodgement forces of all test runs.**
(CSV)

**S1 Fig. Distance force diagrams of all test runs.**
(TIFF)

## Acknowledgments

We would like to thank Benjamin Schuster for his assistance with the study. Furthermore, we thank all manufacturers for providing the CVCs and sutures free of charge.

## Author Contributions

**Conceptualization:** Manuel Florian Struck, Lars Friedrich, Stefan Schleifenbaum, Wolfram Schummer, Bernd E. Winkler.

**Data curation:** Manuel Florian Struck, Lars Friedrich, Stefan Schleifenbaum.

**Formal analysis:** Lars Friedrich, Holger Kirsten.

**Investigation:** Manuel Florian Struck, Lars Friedrich, Stefan Schleifenbaum.

**Methodology:** Wolfram Schummer, Bernd E. Winkler.

**Project administration:** Manuel Florian Struck, Wolfram Schummer, Bernd E. Winkler.

**Resources:** Manuel Florian Struck, Holger Kirsten.

**Software:** Lars Friedrich, Holger Kirsten, Bernd E. Winkler.

**Supervision:** Wolfram Schummer, Bernd E. Winkler.

**Validation:** Lars Friedrich, Holger Kirsten, Bernd E. Winkler.

**Visualization:** Stefan Schleifenbaum, Holger Kirsten.

**Writing – original draft:** Manuel Florian Struck.

**Writing – review & editing:** Lars Friedrich, Stefan Schleifenbaum, Holger Kirsten, Wolfram Schummer, Bernd E. Winkler.

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
