## [Decision Letter · Decision Letter 0]

10 Jul 2019

PONE-D-19-17335

Effectiveness of different central venous catheter fixation suture techniques: an in vitro crossover study

PLOS ONE

Dear Dr. Struck,

Thank you for submitting your manuscript to PLOS ONE. After careful consideration, we feel that it has merit but does not fully meet PLOS ONE’s publication criteria as it currently stands. Therefore, we invite you to submit a revised version of the manuscript that addresses the points raised during the review process.

The article will be reconsidered if the authors will satisfactory address reviewers' concerns.

We would appreciate receiving your revised manuscript by Aug 24 2019 11:59PM. To enhance the reproducibility of your results, we recommend that if applicable you deposit your laboratory protocols in protocols.io, where a protocol can be assigned its own identifier (DOI) such that it can be cited independently in the future. For instructions see: http://journals.plos.org/plosone/s/submission-guidelines#loc-laboratory-protocols

We look forward to receiving your revised manuscript.

Kind regards,

Prof. Raffaele Serra, M.D., Ph.D

Academic Editor

PLOS ONE

Journal Requirements:

2.

We note that you have indicated that data from this study are available upon request. PLOS only allows data to be available upon request if there are legal or ethical restrictions on sharing data publicly. For information on unacceptable data access restrictions, please see http://journals.plos.org/plosone/s/data-availability#loc-unacceptable-data-access-restrictions.

Additional Editor Comments (if provided):

The article is novel and interesting. Please address reviewers' comments.

Reviewers' comments:

Reviewer's Responses to Questions

**Comments to the Author**

1. Is the manuscript technically sound, and do the data support the conclusions?

Reviewer #1: Yes

2. Has the statistical analysis been performed appropriately and rigorously? 

Reviewer #1: Yes

3. Have the authors made all data underlying the findings in their manuscript fully available?

Reviewer #1: No

4. Is the manuscript presented in an intelligible fashion and written in standard English?

Reviewer #1: Yes

5. Review Comments to the Author

Reviewer #1: The authors present an in vitro study comparing the effects of four suturing techniques on maximum dislodgment forces in CVCs from three different manufacturers. The authors conclude that the „finger trap“ fixation needs the highest force to dislodge the CVC.

* The methods section should be condensed. The first sentence (p 4, l 67) can be discarded since it just repeats the last sentence of the introduction. Most of the sub-headers can be removed. The information written in the paragraph for ‚test setting‘ can be mentioned in other parts of the methods.

* The results state a lot of significances but it is not said what these significance stay for. Please, rather state the difference of the statistical significant findings (I.e. high or lower dislodgment rate).

* Is Fig. 5 really important to understand the manuscript? It is only mentioned with one sentence in the results without explaining what this figure is telling us. Please, elaborate or discard.

* The authors write in their conclusion that the combination of the techniques “clamp only” plus “finger trap” might be a reasonable alternative. Although it is reasonable to discuss this it should not be mentioned in the conclusion since this combination was not tested in the study.

## Minor comments

* Protected names should be marked with a trademark sign

* page 4, lines 61-64: in an in vitro study ‚aim‘ instead of ‚outcome‘ would be the more suitable term.

* page 7, line 139: ratios are usually given as part of 1 instead of percentage. It is not clear what kind of ratio you mean (what is your reference of the ratio?)

* Please, add number of measurements for each column as well as results of statistical tests in Table 1

6. PLOS authors have the option to publish the peer review history of their article (what does this mean?). If published, this will include your full peer review and any attached files.

Reviewer #1: Yes: Frank Bloos

---

## [Author Response · Author response to Decision Letter 0]

5 Aug 2019

Response to reviewers comments

Reviewer #1: The authors present an in vitro study comparing the effects of four suturing techniques on maximum dislodgment forces in CVCs from three different manufacturers. The authors conclude that the „finger trap“ fixation needs the highest force to dislodge the CVC.

* The methods section should be condensed. The first sentence (p 4, l 67) can be discarded since it just repeats the last sentence of the introduction. Most of the sub-headers can be removed. The information written in the paragraph for ‚test setting‘ can be mentioned in other parts of the methods.

RESPONSE: 

We would like to thank the reviewer for his comments. We have changed all items as suggested.

* The results state a lot of significances but it is not said what these significance stay for. Please, rather state the difference of the statistical significant findings (I.e. high or lower dislodgment rate).

RESPONSE.

We have completely rewritten this paragraph for clarity. The results section now provides detailed information which dislodgement forces were higher or lower, e.g. “finger trap” and “complete” also resulted in higher dislodgement forces than the “compression” technique.

* Is Fig. 5 really important to understand the manuscript? It is only mentioned with one sentence in the results without explaining what this figure is telling us. Please, elaborate or discard.

RESPONSE:

We have added some more linking information regarding Fig. 5. From our point of view, Fig. 5 is essential for the manuscript because it presents the detailed dislodgement forces for all four fixation techniques and all three manufacturers. Utilizing Fig. 5, the reader can easily see that dislodgement forces are tremendously higher using the “finger trap” or “complete” approach compared to “clamp only” or “compression”. Thus, Fig. 5 helps to demonstrate the benefit of “finger-trap” based approaches at a glance.

* The authors write in their conclusion that the combination of the techniques “clamp only” plus “finger trap” might be a reasonable alternative. Although it is reasonable to discuss this it should not be mentioned in the conclusion since this combination was not tested in the study.

RESPONSE:

We have removed this information from the conclusion part and provided it in the discussion part instead.

## Minor comments

* Protected names should be marked with a trademark sign

RESPONSE:

We have provided all protected brand names either with ® or ™ signs due to the information provided by the manufacturers. 

Furthermore, we have added in the Methods section: “For the sake of simplicity, the terms “Arrow”, “B. Braun” and “Vygon” will be used consistently below.”

* page 4, lines 61-64: in an in vitro study ‚aim‘ instead of ‚outcome‘ would be the more suitable term.

RESPONSE:

We have changed this term as suggested.

* page 7, line 139: ratios are usually given as part of 1 instead of percentage. It is not clear what kind of ratio you mean (what is your reference of the ratio?)

RESPONSE:

We have changed this part as suggested. The information is now presented more precisely and percentages have been replaced by parts of 1. 

* Please, add number of measurements for each column as well as results of statistical tests in Table 1

RESPONSE:

We have now provided the number of measurements for each columns and the statistical results in the legend of Table 1.

We would like to thank the reviewer again for his constructive comments and hope that we have addressed all items appropriately.

We have added another supplementary file including the raw data of our measurements (S1_Dataset) in order to comply with the Journals data sharing policy.

---

## [Decision Letter · Decision Letter 1]

30 Aug 2019

[EXSCINDED]

Effectiveness of different central venous catheter fixation suture techniques: an in vitro crossover study

PONE-D-19-17335R1

Dear Dr. Struck,

We are pleased to inform you that your manuscript has been judged scientifically suitable for publication and will be formally accepted for publication once it complies with all outstanding technical requirements.

With kind regards,

Prof. Raffaele Serra, M.D., Ph.D

Academic Editor

PLOS ONE

Additional Editor Comments (optional):

amended manuscript is acceptable

Reviewers' comments:

Reviewer's Responses to Questions

**Comments to the Author**

1. If the authors have adequately addressed your comments raised in a previous round of review and you feel that this manuscript is now acceptable for publication, you may indicate that here to bypass the “Comments to the Author” section, enter your conflict of interest statement in the “Confidential to Editor” section, and submit your "Accept" recommendation.

Reviewer #1: All comments have been addressed

2. Is the manuscript technically sound, and do the data support the conclusions?

Reviewer #1: Yes

3. Has the statistical analysis been performed appropriately and rigorously? 

Reviewer #1: Yes

4. Have the authors made all data underlying the findings in their manuscript fully available?

Reviewer #1: Yes

5. Is the manuscript presented in an intelligible fashion and written in standard English?

Reviewer #1: Yes

6. Review Comments to the Author

Reviewer #1: (No Response)

7. PLOS authors have the option to publish the peer review history of their article (what does this mean?). If published, this will include your full peer review and any attached files.

Reviewer #1: Yes: Frank Bloos

---

## [Editor Report · Acceptance letter]

5 Sep 2019

PONE-D-19-17335R1 

Effectiveness of different central venous catheter fixation suture techniques: an in vitro crossover study 

Dear Dr. Struck:

I am pleased to inform you that your manuscript has been deemed suitable for publication in PLOS ONE. Congratulations! Your manuscript is now with our production department. 

With kind regards,

on behalf of

Prof. Raffaele Serra 

Academic Editor

PLOS ONE